# Acute and Sub-Chronic Exposure to Artificial Sweeteners at the Highest Environmentally Relevant Concentration Induce Less Cardiovascular Physiology Alterations in Zebrafish Larvae

**DOI:** 10.3390/biology10060548

**Published:** 2021-06-18

**Authors:** Ferry Saputra, Yu-Heng Lai, Rey Arturo T. Fernandez, Allan Patrick G. Macabeo, Hong-Thih Lai, Jong-Chin Huang, Chung-Der Hsiao

**Affiliations:** 1Department of Bioscience Technology, Chung Yuan Christian University, Taoyuan 320314, Taiwan; ferrysaputratj@gmail.com; 2Department of Chemistry, Chinese Culture University, Taipei 11114, Taiwan; LYH21@ulive.pccu.edu.tw; 3Laboratory for Organic Reactivity, Discovery and Synthesis (LORDS), Research Center for the Natural and Applied Sciences, University of Santo Tomas, Espana St., Manila 1015, Philippines; reyarturo.tapia.fernandez@gmail.com (R.A.T.F.); agmacabeo@ust.edu.ph (A.P.G.M.); 4Department of Aquatic Biosciences, National Chiayi University, Chiayi 600355, Taiwan; 5Department of Applied Chemistry, National Pingtung University, Pingtung 90003, Taiwan; 6Center for Nanotechnology, Chung Yuan Christian University, Taoyuan 320314, Taiwan; 7Research Center for Aquatic Toxicology and Pharmacology, Chung Yuan Christian University, Taoyuan 320314, Taiwan

**Keywords:** artificial sweeteners, zebrafish, cardiac performance

## Abstract

**Simple Summary:**

The usage of artificial sweetener has been increased from year to year as the result of pursuing healthy lifestyle. However, ironically, several studies suggest that the consumption of artificial sweeteners cause sugar-related adverse effects (e.g., obesity, type 2 diabetes and cardiovascular disease). In this study, we explore the potential cardiovascular adverse effect of several artificial sweeteners using zebrafish as animal model. We found that artificial sweetener at the highest concentration found in nature only slightly alter the cardiovascular performance of zebrafish larvae. Furthermore, no alteration of cardiac performance showed after longer incubation which support the safety of artificial sweeteners.

**Abstract:**

Artificial sweeteners are widely used food ingredients in beverages and drinks to lower calorie intake which in turn helps prevent lifestyle diseases such as obesity. However, as their popularity has increased, the release of artificial sweetener to the aquatic environment has also increased at a tremendous rate. Thus, our study aims to systematically explore the potential cardiovascular physiology alterations caused by eight commercial artificial sweeteners, including acesulfame-K, alitame, aspartame, sodium cyclamate, dulcin, neotame, saccharine and sucralose, at the highest environmentally relevant concentration on cardiovascular performance using zebrafish (*Danio rerio*) as a model system. Embryonic zebrafish were exposed to the eight artificial sweeteners at 100 ppb and their cardiovascular performance (heart rate, ejection fraction, fractional shortening, stroke volume, cardiac output, heartbeat variability, and blood flow velocity) was measured and compared. Overall, our finding supports the safety of artificial sweetener exposure. However, several finding like a significant increase in the heart rate and heart rate variability after incubation in several artificial sweeteners are noteworthy. Biomarker testing also revealed that saccharine significantly increase the dopamine level in zebrafish larvae, which is might be the reason for the cardiac physiology changes observed after saccharine exposure.

## 1. Introduction

Artificial sweeteners or non-nutritive sweeteners are widely consumed as food additives due to their non-existent caloric content and sweeter taste than normal sucrose. They are mainly used in beverages and other diet products as sugar substitutes by consumers who seek a healthier lifestyle. With the increased popularity of non-caloric sweetener usage and production, an increase in the amount of artificial sweeteners released into the aqueous environment has also been confirmed from year to year [1]. Based on several previous studies, several artificial sweeteners, e.g., saccharine, sodium cyclamate, neotame, acesulfame K, and sucralose were found in various aquatic environments at concentrations ranging from 3.5 ng/L to 0.12 mg/L, among which, according to Gan et al., sodium cyclamate has the highest concentration in the aquatic environment [2,3,4,5]. Artificial sweeteners can get into the water cycle not only from industrial waste, but also from household and even human excretion that enters into wastewater treatment plants through which in most cases they pass without any changes. Furthermore, their high solubility in water and resistance to degradation in nature make them emerge as persistent pollutants in the aquatic environment [6].

Artificial sweeteners from the aquatic environment could easily come back to humans. Aside from the possibility of bioaccumulation by planktonic animals, the practice of drinking water directly from the water source is widespread, especially in underdeveloped countries [7]. Some studies even report that artificial sweeteners were detected in tap water [6,8]. This might lead to a risk of accumulation in the human body and cause some adverse effect because contrary to their designation as non-caloric alternatives to sugar, artificial sweeteners have ironically been linked to the consumption of more calories due to their ability to increase sugar cravings and dependence as well as impairing caloric compensation leading to appetite stimulation [9]. Independent from this mechanism, artificial sweeteners are associated with impaired glucose tolerance secondary to altered gut microbiota [10,11]. These are some of the proposed mechanisms that despite artificial sweeteners being non-caloric are associated with obesity, type 2 diabetes mellitus, and cardiovascular diseases [12]. These associated life-style related diseases are clearly linked to the development of atherosclerotic plaques which may cause ischemia and acute coronary syndrome [13,14]. However, we hypothesized that artificial sweeteners may also directly affect cardiac function and cause damage independent of plaque formation. This is supported by studies done on Wistar albino rats wherein aspartame induced oxidative stress in cardiac muscle and increased heart rate variability [15,16]. A study done in aquatic animals also showed that aspartame could cause oxidative stress in the brain, gills, and muscles of common carp [5]. Related diseases and published effects of artificial sweetener exposure in humans and some animal models are listed below (Table 1).

Understanding the mechanism of artificial sweeteners on cardiovascular disease requires models where the parameters of interest are easily measured but at the same time can represent the complexity of the human heart. *Danio rerio*, also known as the zebrafish, is a commonly used vertebrate model in physiologic, genetic, and regenerative experiments on cardiac diseases [29,30,31,32]. Among the vertebrate species, it is particularly favored because of its relatively short life span which allows investigators to monitor the disease in an accelerated phase [33]. 

Like in humans, the cardiac cycle in zebrafish was also divided into two phases, systole (contraction) and diastole (relaxation) [34]. Diastole is affected by decreasing the time that it takes for blood to fill the ventricles. Conversely, at slower heart rates, left ventricular end-diastolic volume is larger. It is also during diastole that blood flows to the coronary arteries to supply the heart [35]. Increases in heart rate will decrease coronary perfusion time, which was observed in patients with coronary stenosis and leads to tissue ischemia. Furthermore, the elevation of heart rate adversely increases the demand for myocardial oxygen [36]. Increases in heart rate also adversely cause pulmonary hypertension since it worsens biventricular function [37]. Hence, artificial sweeteners that cause an increase in heart rate should not be given to patients with coronary artery stenosis for it will further lead to tissue ischemia.

In the context of our interest in providing in vivo models to study the effects of substances in a variety of disease model systems [38,39,40], zebrafish were explored as an animal model. In this study, we investigated the possibility of cardiovascular performance alteration after exposure to eight different artificial sweeteners at 100 ppb concentration, which is highest environmentally relevant concentration of artificial sweeteners according to the Gan et al. report [2], by measuring the blood flow velocity and different cardiac physiology parameters such as the stroke volume, cardiac output, ejection fraction, and shortening fraction, along with heart rate, and heart rate variability as the change in cardiovascular physiology is one of the indexes of cardiotoxicity [41]. Furthermore, we also check several neurotransmitters related to stress and oxidative stress as those were related to the changes in cardiac physiology [42,43].

## 2. Materials and Methods 

### 2.1. Animal Ethics and Artificial Sweetener Exposure

All experiments involving zebrafish were performed following the guidelines approved by the Institutional Animal Care and Use Committees (IACUCs) of the Chung Yuan Christian University (Approval No. 109001, issue date 15 January 2020). In this study, wild type AB strain zebrafish was used as a vertebrate model and maintained in a continuously filtered and aerated water system. The temperature was maintained at 26 ± 1 °C with 14/10 h of light/dark cycle according to previously reported protocols [44]. After the embryos were collected, they were reared in an incubator at 28 ± 1 °C until the time of treatment.

In this study, eight artificial sweeteners (acesulfame K, alitame, aspartame, dulcin, neotame, saccharine, sodium cyclamate, and sucralose) were tested for their potential cardiovascular physiology alteration. All of the artificial sweeteners were purchased from Aladdin Chemicals (Shanghai, China). Stock solutions of artificial sweeteners were prepared with distilled water. During assays, the stock solutions were diluted to a testing concentration of 100 ppb and applied to embryonic zebrafish (Table 2). MS222 (tricaine methanesulfonate) with the concentration of 100 ppm and glucose with the concentration of 100 ppb were used as the controls. MS222 has been used as an anesthetic for zebrafish and is known to decrease cardiac performance [45,46], while glucose is known to increase cardiac parameters [47,48]. 

For acute exposure, zebrafish larvae at 60 h post fertilization (hpf) was exposed to various artificial sweeteners for 12 h and at 72 hpf the cardiovascular system was analyzed. For the sub-chronic exposure, the larvae was already exposed to artificial sweetener at day1 and at day 8, the cardiovascular system was measured. The water was change every 2 days to keep the medium fresh. A schematic diagram that shows the detailed experimental design can be seen in Figure 1. The experiments were done in triplicate with an average of five individuals per compound for each replication.

### 2.2. Cardiovascular Performance Measurement

To record zebrafish heartbeats and cardiac physiology parameters, 3% methylcellulose was used as a mounting agent to minimize zebrafish movement during video recording. A high-speed digital charged coupling device (CCD) (AZ Instruments, Taichung, Taiwan) was mounted on an inverted microscope (Sunny Optical Technology, Yuyao, China) to record zebrafish heartbeat and blood flow. To acquire better image contrasts and resolutions, Hoffmann objective lenses with 40× magnification were used, and the video was recorded at 200 frames per second (fps) for 10 s according to our previously published protocol [49]. To calculate the blood flow velocity of zebrafish, the “Trackmate” plug-in in the ImageJ software was used, while heart rate analysis was done using the Time Series Analyzer V3 plug-in (https://imagej.nih.gov/ij/plugins/time-series.html (accessed on 15 June 2021)) to analyze the pattern of changes in dynamic pixel intensity [50,51]. Heart rate, expressed as beats per minute (bpm), was measured using the Peak analyzer function in OriginPro 2019 software (Originlab Corporation, Northampton, MA, USA) by determining the time interval of each peak. The Poincare plot was generated using a Poincare plug-in from the OriginPro 2019 software. Sd1 and sd2 extracted from the plots were recorded and statistically analyzed to calculate heart rate. Stroke volume is determined by the assumption that the heart chamber has an ellipsoid shape and is calculated by subtracting end-systolic volume (ESV) from the end-diastolic volume (EDV). The volume of the heart chamber was calculated using the heart chamber long (D_L_) and short-axis (D_S_) to compare the volume difference between EDV and ESV [52]. Cardiac output was calculated by multiplying the heart rate observed in the ventricle with stroke volume. Ejection fraction was likewise calculated by dividing stroke volume with EDV while shortening fraction was determined by obtaining the ratio of the length of the heart chamber at the end of the systolic phase to the end of a diastolic phase, which represents muscular contractility of the heart [53]. Ejection fraction and shortening fraction was calculated using the following formulas:$$EF\left(\%\right)=\frac{SV}{EDV}\times 100\%\;$$
$$SF\left(\%\right)=\frac{Ds\left(EDV\right)-Ds\left(ESV\right)}{Ds\left(EDV\right)}\times 100\%$$

### 2.3. Determination of Biomarker Content

Live embryos were collected at 72 hpf and four neurotransmitters or biomarkers related to stress and oxidative stress like acetylcholine esterase (AChE), dopamine, cortisol, and reactive oxygen species (ROS) were selected and their relative contents in the whole larvae was calculated using commercial kits according to the manufacturer’s instructions (Zgenebio Inc., Taipei, Taiwan). The whole-body of larvae was minced and homogenized in phosphate buffer saline (PBS) using a tissue homogenizer and the total protein was collected and calculated using BCA™ Protein Assay Kit (Thermo Scientific, Rockford, IL, USA) according to the manufacturer’s instructions. The experiment was done with five replicates per sample (*n* = 30/sample).

### 2.4. Biostatistics

Statistical analysis was done using GraphPad Prism (GraphPad Inc., La Jolla, CA, USA). Normality test and relative standard deviation measurement was done before statistical test in order to select statistical test. Either parametric or non-parametric ANOVA tests were performed according to the normality of data distribution and the variance of the data and the significance was calculated based on the appropriate post-hoc multiple comparison test.

## 3. Results

### 3.1. Cardiac Performance in Zebrafish after Acutely Exposed to Artificial Sweeteners

None of the artificial sweeteners tested in this study induced any embryonic deaths or cardiac deformation (like edema) in zebrafish (Figure A1 in Appendix A). Next, we asked whether artificial sweetener exposure can induce cardiac performance alteration. Several important parameters like blood flow, heart rate, cardiac interval, stroke volume, cardiac output, ejection fraction, and shortening fraction were analyzed by using our previously established ImageJ-based method [49,50]. Heart rate variability was calculated using a Poincare plot to check cardiac rhythm after exposure to the different artificial sweeteners.

To measure the amount of blood pumped during for each systolic phase, stroke volume was determined. No significant change was observed in zebrafish stroke volume after incubation in low dose 100 ppb of all artificial sweeteners (*p* > 0.05) (Figure 2A). Similar result also observed in cardiac output data as no significant change observed in 100 ppb concentration compared to the non-treated group (*p* > 0.05) (Figure 2B). 

Heart rate can be affected by hormones, temperature, or by exogenous compounds [54]. Low time intervals, as noted in increased heart rate, resulting in a reduced stroke volume secondary to a decrease in the ventricular filling. Humans who train with high-intensity exercise, show manifest longer time intervals between each beat that allows the ventricle to fill with blood efficiently [55]. In this study, we observed that acesulfame K (177.7 ± 12.83 beats per minute (BPM), *p* = 0.0048), neotame (175.8 ± 10.86 BPM, *p* = 0.0217), saccharine (176 ± 9.27 BPM, *p* = 0.0223), and sucralose (177.4 ± 13.9 BPM, *p* = 0.0059) exposure significantly increased zebrafish heart rate compared to the non-treated group (Figure 2C). 

Ejection fraction and shortening fraction was analyzed to determine the heart muscular contractility [53]. No significant change in the ejection fraction was observed among the different artificial sweeteners when used on zebrafish at 100 ppb concentration compared to the non-treated group (*p* > 0.05) (Figure 2D). However, a significant decrease was observed in zebrafish shortening fraction after incubation with sodium cyclamate (12.53 ± 4.999%, *p* = 0.0101) (Figure 2E). 

### 3.2. Heart Rate Regularity in Zebrafish after Acute Exposure to Artificial Sweeteners

Using the Poincare plots, the artificial sweeteners were checked to see whether they cause alterations in the heart rate variability. A Poincare plot is a method that plots the heartbeat interval between two successive heartbeats and has been used to study heart rate variability in humans, rodents, and zebrafishes [56,57,58,59]. A higher standard deviation means that the heartbeat is more irregular. In this study, significant increment was observed after incubation in dulcin (0.01985 ± 0.0082 s, *p* = 0.034), saccharine (0.02251 ± 0.01198 s, *p* = 0.0158), and sodium cyclamate (0.02197 ± 0.01372 s, *p* = 0.0426) at sd1 but showed no significant difference with sd2 compared to non-treated group (Figure 3A,B). 

### 3.3. Blood Flow Velocity of Zebrafish after Exposure to Artificial Sweeteners

To further check the effect of artificial sweeteners on the cardiovascular system, blood flow measurements were performed. Blood flow velocity is related to the ability of individuals’ heart contractility. Furthermore, by measuring blood flow velocity, other health conditions like stress conditions also can be measured [49,60]. In our previous published method, we found that the maximal and average blood flow rate showed more dynamic changes after challenging with chemicals in the dorsal aorta of zebrafish embryos [49]. In this test, we measured the maximal and average blood flow rate in zebrafish embryos after acute exposure to artificial sweeteners. In line with the cardiac output data, no significant change was observed in embryos after artificial sweetener exposure in terms of maximum and average blood flow velocity (*p* > 0.05) (Figure 4), which suggests that artificial sweeteners at highest the environmentally relevant concentration do not have any significant effect on the vascular system of zebrafish.

### 3.4. Comparison of Stress and Oxidative Stress-Related Biomarkers in Zebrafish after Exposure to Artificial Sweeteners

In this study, the possibility of the alteration in neurotransmitters or biomarkers related to stress was done by checking the acetylcholine esterase (AChE), cortisol, and dopamine levels in whole fish lysates. The relative oxidative stress level was checked by measuring reactive oxygen species (ROS). After incubation in artificial sweeteners for 12 h, we found that saccharine significantly increased the dopamine level (13.08 ± 3.438 pg/mg, *p* = 0.0466) in zebrafish compared to the control group (Figure 5C). However, no significant change was observed in the other biomarkers tested (AChE, cortisol and ROS) (Figure 5A,B,D) (*p* > 0.05), which suggests that acute artificial sweetener exposure does not cause stress to zebrafish larvae.

### 3.5. Cardiovascular Performance of Zebrafish Larvae after Sub-Chronic Exposure of Artificial Sweetener

After the acute exposure data showed that artificial sweeteners have very little effect on the cardiovascular system, a follow-up experiment was conducted to validate the effect of sub-chronic incubation on the zebrafish cardiovascular system as the adverse effects of artificial sweeteners usually become apparent after prolonged consumption. Interestingly, the significant increment observed in several cardiac performance parameters after acute exposure was nowhere to be observed after sub-chronic exposure (*p* > 0.005) (Figure 6 and Figure 7). However, no data regarding the heart rate regularity can be extracted in sub-chronic incubation as the fish are already strong enough to make a vibrating movement inside the mounting solution, and thus can compromise the heart rate regularity data. These data suggest that both acute and sub-chronic incubation with various artificial sweetener at the highest environmental relevant concentration has very little effect on the cardiovascular performance of zebrafish larvae. 

## 4. Discussion

The effects of a number of substances in aquatic model animals such as zebrafish, medaka, *Daphnia*, and *Xenopus* have long been investigated [61,62,63,64,65]. In order to address the possible physiological alterations caused by commercial artificial sweeteners, we used zebrafish cardiovascular physiology as a target, since it has been recognized as an excellent and sensitive model for toxicity assessment [66,67,68]. After acute incubation with artificial sweeteners, we measured several cardiovascular physiology endpoints and neurotransmitters/biomarkers since that change is tightly associated with cardiac physiology and play as an important index for cardiotoxicity assessment [41]. 

The key utility of this paper is to provide solid evidence to support the notion that in vivo acute exposure to artificial sweeteners at the highest environmentally relevant concentration causes very small degree of alteration on the zebrafish cardiovascular system. One possible explanation for this observation may be associated with the metabolic fate of artificial sweeteners. By nature, artificial sweeteners are not readily absorbed by the body and most of them will be discharged in the urine and sweat which makes them have a minimal effect on the body [25,69]. More information about the metabolic fate of artificial sweeteners in humans is summarized in Table 3. Another interesting issue for artificial sweeteners is the exposure time. In humans adverse effects of artificial sweeteners indeed can be found after chronic exposure [70,71,72] which lead us to perform follow-up sub-chronic exposure tests. Although our data suggest that no significant difference was observed, it will be worth to note that the potential chronic toxicity of artificial sweetener exposure cannot be ignored as some research suggest that artificial sweetener could accumulate inside the body. A study has reported sucralose accumulation in blood plasma after administration of water with added sucralose or diet soda containing sucralose [73]. Also, more sucralose accumulation than of acesulfame K in the bile and gills has been reported in *Sparus aurata* [74]. In addition, a study has suggested the accumulation of sucralose in the tissue of zebrafish after two hours of exposure [75]. To address this speculation, in future studies experiments using isotope labeling or chemically modified artificial sweeteners in the zebrafish system could provide more direct evidence. 

In this study, we observed that acesulfame K, neotame, saccharine and sucralose induced some increase in zebrafish heart rate. Although in this study no significant change in ROS level after artificial sweetener exposure was observed, a previous study showed that sucralose treatment caused oxidative stress in the brain by increasing malondialdehyde (MDA) levels and decreasing neuron cells in rat [79]. Studies by Crus-Rojas et al. and Saucedo-Vence et al. also reported that acesulfame K and sucralose increase the oxidative activity in the brain of common carp [5,80]. Heart rate is controlled by signals from the brain, therefore, damage in the brain may cause alterations in heart rate regulation [81,82,83]. A previous study in common carp also suggested lower antioxidant capacity after sucralose exposure [84]. A decrease in antioxidant and an increase in catalase is a marker for ROS levels, which can cause tissue damage, especially in sensitive organs such as the heart. 

Cardiac function is mediated by the sympathetic and parasympathetic nervous systems [85]. The sympathetic nervous system acts on adrenergic receptors (AR) which are G-protein-coupled receptors (GPCR) [86]. Stimulation on β_1_ARs and β_2_ARs in the heart increases cardiac contractility, frequency, rate of relaxation, acceleration of impulse conduction through the atrioventricular node as well as increased pacemaker activity from the sinoatrial node by increasing intracellular Ca^2+^ concentration [87]. Previous studies reported that aspartame can increase brain adrenergic neurotransmitters in various parts of the mouse brain [88]. Furthermore, aspartame increases sympathetic activity within half an hour after consumption either in the form of diluted water or in aspartame-sweetened diet drinks in humans [89]. These findings suggest the possibility of artificial sweeteners altering the sympathetic nervous system and finally inducing cardiac performance alterations in zebrafish.

After incubation with saccharine, the heart rate and dopamine level were significantly elevated in zebrafish. Dopamine is one of the hormones that acts as a neurotransmitter, which functions primarily in the central nervous system and is usually related to happiness. The role of dopamine in mediating food reward and stimulating palatability is well-established. Sucrose induces dopamine release in rats [90]. Also, the uptake of saccharine increases dopamine levels in rats which similar to our finding in this study [91]. Among the reasons for the increase of dopamine release is related to the food reward system that is induced by sweet taste [92]. The positive effect of dopamine on heart rate and muscle contractility has been observed in animal models, such as dogs [93] and rats [94], thus corroborating our results that the increase in the heart rate was due to the increase of dopamine level induced by those artificial sweeteners.

T1R and T2R are receptors expressed on taste buds that belong to a superfamily of GPCRs mediating sweet stimuli in humans and are highly expressed in the olfactory system, especially in the tongue [95,96]. Those receptors have also been isolated and characterized in fish as well, which share high conservation with human T1R and T2R counterparts [97]. T1R homolog in zebrafish responds to artificial sweeteners by increasing dopamine concentration [98]. Therefore, we proposed this might be one of the reasons explaining why some artificial sweeteners increase zebrafish dopamine levels.

Although acesulfame K, neotame, and sucralose exposure at 100 ppb caused an increment in heart rate, no change in biomarker level was detected for those artificial sweeteners. Aside from the trait of acesulfame K as sweetener, a persistent bitter aftertaste was also noted as the trait of this sweetener and it also increases as the concentration increases [99,100,101]. While sweetness increases dopamine release via the food reward system, aversive bitterness can decrease the release of dopamine [102]. Furthermore, previous studies also suggest that the sweetness potency of some sweeteners will decrease as the sweetener level increases, which might be the reason for the fact no alteration was observed with acesulfame K and neotame [103,104,105]. An epidemiology study collected by Dietrich et al. also showed that saccharine has a lower taste threshold compared to the other three sweeteners, which might be the reason for the dopamine level increment [6].

Another interesting finding was the fact that the significant difference in cardiac performance parameters observed after acute exposure was not shown after sub-chronic exposure. This might be related to the increase of leptin after sub-chronic incubation. Leptin is a hormone that regulates the sensitivity of sweet receptors present in oral cavity via the leptin receptor and an increase of leptin concentration will reduce the sweet sensitivity. It is normally produced in adipose tissue and creates a negative feedback loop for sweet taste stimuli [106]. A previous study by Sigala et al. showed that the consumption of aspartame and sucrose caused some increase in plasma leptin levels after 2 weeks, which might be the reason why a significant increment of cardiac performance parameter was not observed after chronic incubation [107]. 

γ-Aminobutyric acid (GABA) is a neurotransmitter that plays an important role in inhibiting neuronal activity. In vertebrates, including zebrafish, the heart rate is controlled by GABA signaling [108,109]. For example, the activity of the dopamine D_2_ receptor modulated AKT signaling and altered GABAergic neuron development and motor behavior in zebrafish larvae [110]. The addition of dopamine significantly increased the variability of sd1 in zebrafish larvae [111]. Therefore, we hypothesized the heart rate in both zebrafish is mediated by the GABA system, and the administration of artificial sweeteners can trigger similar heart rate variability in zebrafish larvae.

Although no significant change was observed in the AChE and cortisol levels in zebrafish after artificial sweetener exposure, those two biomarkers are biomarkers that are related to stress which has a direct impact on heart rate [112,113,114]. Acetylcholinesterase is an enzyme that will break down acetylcholine into acetic acid and choline which is primarily found in neuromuscular junctions. Acetylcholine usually is released under stress conditions and binds to the M2 muscarinic receptor to decrease the heart rate, thus an increase in AChE levels will cause a lower acetylcholine level which makes the heart rate consistently be at a higher rate compared to normal conditions under stress [115]. An increase in cortisol level is a response to stress and will increase the heart rate and blood pressure which if the condition is prolonged, will result in various cardiovascular-related deseases [116]. 

## 5. Conclusions

Our study aimed to elucidate the physiological effects of artificial sweetener exposure at high environmentally relevant concentrations on the cardiovascular system of zebrafish larvae. The absence of significant phenotypic changes (like edema) during the experiments and minimal effects on the cardiovascular system after both acute and sub-chronic artificial sweetener exposure support the safety of artificial sweeteners for zebrafish larvae. The result of the biomarker assays showed that the cardiac physiology alteration observed after saccharine exposure is associated with dopamine content elevation. The overall biomarker test did not favor stress elevation in zebrafish after 12 h incubation of artificial sweeteners. In the future, chronic exposure experiments are considered necessary and important to be done as follow-up experiments in order to fully recapitulate the potential biological effect of artificial sweeteners in zebrafish system.

## Figures and Tables

**Figure 1 biology-10-00548-f001:**
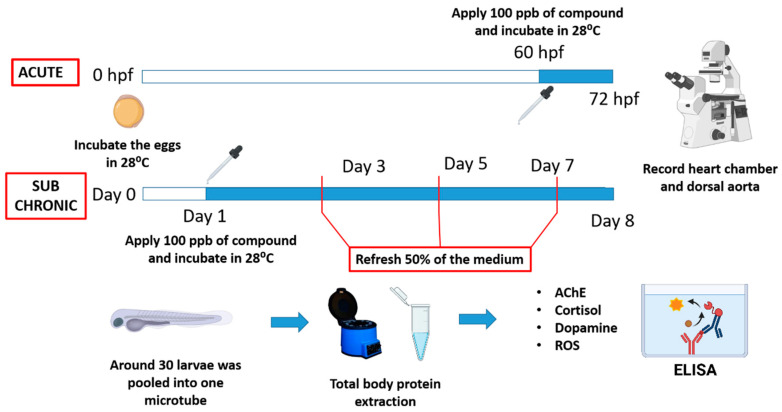
Schematic diagram showing our experimental design for testing cardiovascular physiology (upper panel) and biomarker alteration (lower panel) caused by eight artificial sweeteners in zebrafish embryos. hpf, hour post-fertilization; AChE, acetylcholine esterase; ROS, reactive oxygen species; ELISA, enzyme-linked immunosorbent assay.

**Figure 2 biology-10-00548-f002:**
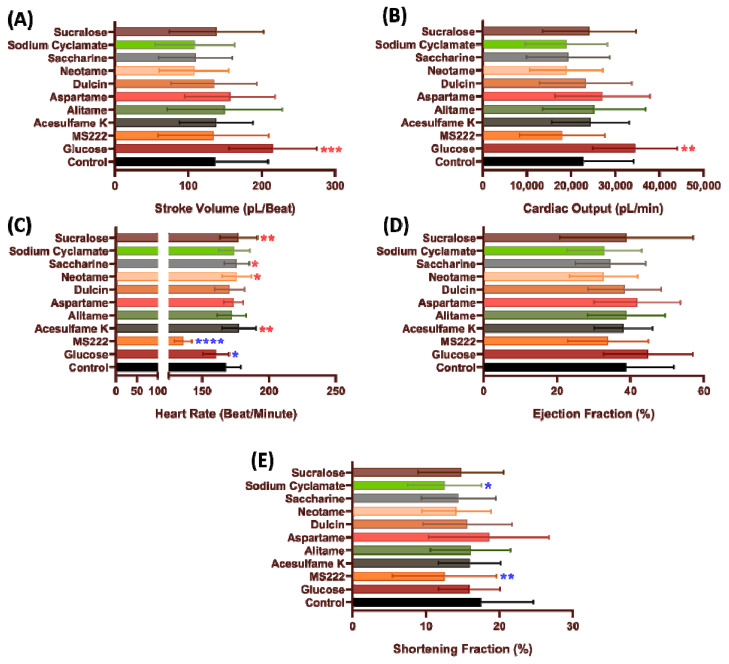
Cardiac performance endpoints for zebrafish after acute exposure to eight different artificial sweeteners at highest environmental relevant concentration (100 ppb). Cardiac physiology endpoints like (**A**) Stroke volume, (**B**) Cardiac output, (**C**) Heart rate, (**D**) Ejection fraction and (**E**) Shortening fraction were measured and compared. Data were presented as mean ± SD and statistical significances were tested by Ordinary One-Way ANOVA test followed by Fisher’s LSD test as post-hoc multiple comparison test (*n* = 15). Red asterisk shows significant increase while blue asterisk shows significant decrease when compared to control. (* *p* < 0.05, ** *p* < 0.01, *** *p* < 0.001, **** *p* < 0.0001).

**Figure 3 biology-10-00548-f003:**
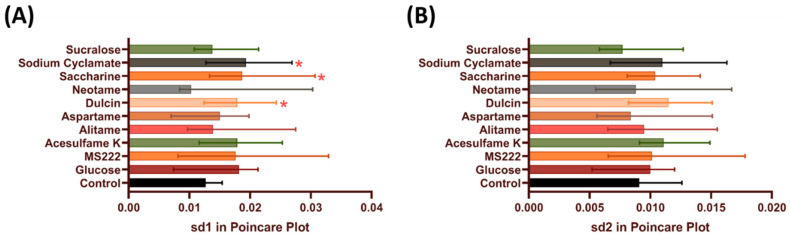
Comparison of heart rate variability in zebrafish larvae after acute incubation of eight different artificial sweeteners at highest environmental relevant concentration (100 ppb). Data analysis was based on the standard deviation 1 (**A**) and standard deviation 2 (**B**) generated by Poincare Plot and presented as median with interquartile range. Statistical significances were tested by Kruskal–Wallis test followed by uncorrected Dunn’s test as post-hoc multiple comparison test (*n* = 15). Red asterisk shows significant increase compared to control (* *p* < 0.05).

**Figure 4 biology-10-00548-f004:**
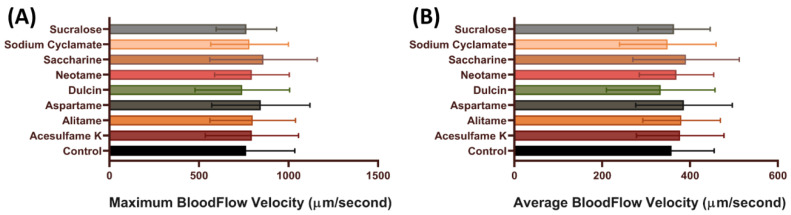
Comparison of maximum (**A**) and average (**B**) blood flow velocity in zebrafish larvae after acute incubation of eight different artificial sweeteners at highest environmental relevant concentration (100 ppb). Data were presented as mean ± SD and statistical significances were tested by Ordinary One-Way ANOVA test followed by Fisher’ LSD test as post-hoc multiple comparison test (*n* = 15).

**Figure 5 biology-10-00548-f005:**
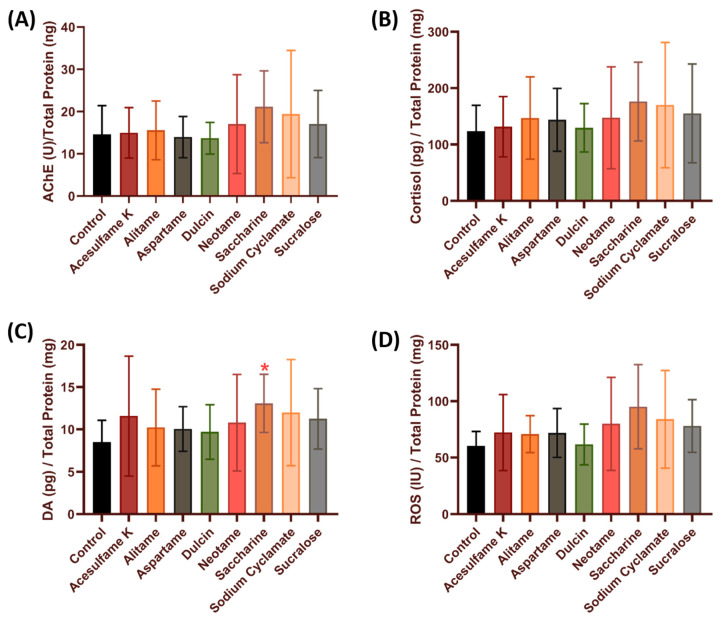
Comparison of neurotransmitter related to stress and oxidative stress in zebrafish embryos after incubation of eight different artificial sweeteners at highest environmental relevant concentration (100 ppb). (**A**) AChE (acetylcholinesterase), (**B**) cortisol, (**C**) dopamine and (**D**) ROS (reactive oxygen species). Data were presented as mean ± SEM and statistical significances were tested by Brown-Forsythe ANOVA test followed by post-hoc unpaired t with Welch’s correction test as multiple comparison test (*n* = 5). Red asterisk shows significant increase (* *p* < 0.05).

**Figure 6 biology-10-00548-f006:**
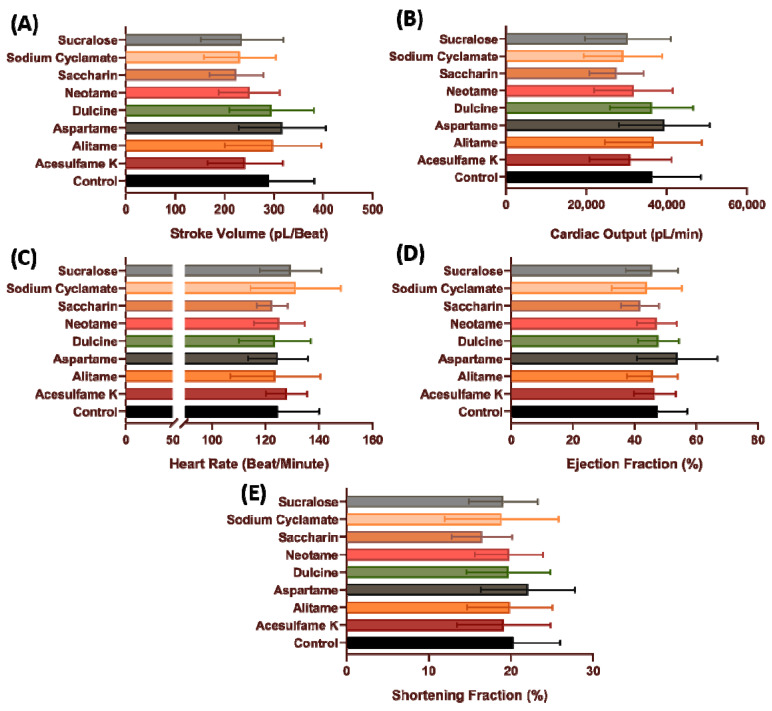
Cardiac performance endpoints for zebrafish after sub-chronic exposure to eight different artificial sweeteners at highest environmental relevant concentration (100 ppb). Cardiac physiology endpoints like (**A**) Stroke volume, (**B**) Cardiac output, (**C**) Heart rate, (**D**) Ejection fraction and (**E**) Shortening fraction were measured and compared. Data were presented as mean ± SD and statistical significances were tested by Ordinary One-Way ANOVA test followed by Fisher’s LSD test as post-hoc multiple comparison test (*n* = 15).

**Figure 7 biology-10-00548-f007:**
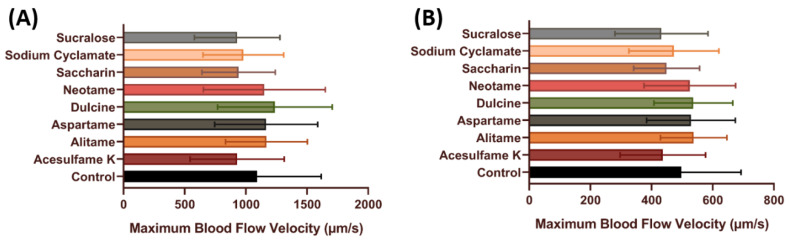
Comparison of maximum (**A**) and average (**B**) blood flow velocity in zebrafish larvae after sub-chronic incubation of eight different artificial sweeteners at highest environmental relevant concentration (100 ppb). Data were presented as mean ± SD and statistical significances were tested by Ordinary One-Way ANOVA test followed by Fisher’s LSD test as post-hoc multiple comparison test (*n* = 15).

**Table 1 biology-10-00548-t001:** Study of related disease and effect on biomarker after artificial sweetener exposure.

	Related Disease	Effect on Biomarker
**Human**	Obesity, diabetes, cardiovascular event (artificial sweetener beverage) [17], hepatotoxicity (saccharine) [18], nausea, vomiting, thrombocytopenia (aspartame) [19,20].	Lactate dehydrogenase ↑, acetylcholinesterase ↓ (sspartame) [21,22].
**Rodent**	Obesity, gut biota community shift (acesulfame K) [11], vascular endothelial dysfunction (acesulfame K & sucralose) [23], glucose intolerance (saccharine) [24], thyroid tumor (acesulfame K) [25] (Rat).	Dopamine ↑, hydroxytryptamine ↑, norepinephrine ↑, epinephrine ↑ (acesulfame K) [26], xanthine oxidase ↑, superoxide dismutase ↑, catalase ↑ (aspartame) [15] (rat).
**Fish**	Swimming defect, inflammatory in brain and liver, growth malformation (aspartame) [27,28] (zebrafish).	Reactive oxygen species ↑ (aspartame) [27] (zebrafish), superoxide dismutase ↑, catalase ↑, lipid peroxidase (sucralose) [5] (common carp).

↑ means an increase and ↓ means a decrease.

**Table 2 biology-10-00548-t002:** Molecular formula and acute toxicity information of the eight artificial sweeteners.

Number	Artificial Sweetener	Molecular Formula	Aquatic Acute Toxicity
1	acesulfame K	C_4_H_4_KNO_4_S	LC_50_: 96 hr for fish: (mg/L): >1000
2	alitame	C_14_H_25_N_3_O_4_S	N.A.
3	aspartame	C_14_H_18_N_2_O_5_	N.A.
4	dulcin	C_9_H_12_N_2_O_2_	N.A.
5	neotame	C_20_H_30_N_2_O_5_	N.A.
6	saccharine	C_7_H_5_NO_3_S	N.A.
7	sodium cyclamate	C_6_H_12_NO_3_SNa	N.A.
8	sucralose	C_12_H_19_Cl_3_O_8_	N.A.

WHO GHS acute aquatic toxicity definition is 96 hr LC_50_ less than 1 ppm for fish or 48 hr EC50 less than 1 ppm for Crustaceans. N.A. not available.

**Table 3 biology-10-00548-t003:** Absorption and metabolic fate of the eight artificial sweeteners used in the study.

Number	Artificial Sweetener	Absorption in Humans	Metabolic Fate in Humans
1	acesulfame K	<1% [76]	Not metabolized [76]
2	alitame	100% [76]	Rapidly metabolized [76]
3	aspartame	100% [69]	Metabolized into methanol, aspartic acid, and phenylalanine [69]
4	dulcin	N.A.	Metabolized into 4-aminophenol [77]
5	neotame	100% [76]	Rapidly metabolized [76]
6	saccharine	0% [69]	Bind to plasma protein and distributed via blood without metabolized [69]
7	sodium cyclamate	Poorly absorbed [78]	Metabolized into cyclohexamine [76]
8	sucralose	<10% [76]	Not metabolized [76]

## Data Availability

The data presented in this study are available directed to the corresponding authors.

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
