# Peer review of "Acute and Sub-Chronic Exposure to Artificial Sweeteners at the Highest Environmentally Relevant Concentration Induce Less Cardiovascular Physiology Alterations in Zebrafish Larvae"

_biology, 2021, doi:10.3390/biology10060548_

Round 1

Reviewer 1 Report

The manuscript was improved; all of my comments were addressed. Please check the new manuscript version because section 2.3 is missing

Author Response

The manuscript was improved; all of my comments were addressed. Please check the new manuscript version because section 2.3 is missing

We thank you for the detailed review for the manuscript. As the reviewer mention, it seems the authors make some mistyping for the section number. Thus, in the updated manuscript, the authors already corrected this problem. Thank you for your carefully reviewing.

Reviewer 2 Report

In this manuscript, Saputra et al., showed the impact of acute and sub chronic exposure to 8 artificial sweeteners in zebrafish cardiac performance, by measuring cardiac physiological parameters. The authors showed that the studied sweeteners have a very little effect on zebrafish larva cardiac performance.

I just have some minor suggestions to the authors to improve the manuscript

Minor concerns:

1. Introduction

1.1.In the first paragraph of the manuscript, the authors start the introduction by describing the presence of artificial sweeteners in aquatic environment explaining their release, referring their concentration and difficulty in their degradation. On the second paragraph, the authors refer the consequences of the usage of artificial sweeteners in human diseases as obesity, diabetes and cardiovascular diseases showing also the data obtained from animal models. The relation between these two topics as well as the relation between the presence of artificial sweeteners in water and the aims of the present manuscript are not clear. I would suggest to the authors to add the missing information to introduction

1.2. In the sentence “ Based on several previous study, artificial sweetener was found in several aquatic environment with concentration ranging from 3.5 ng/L to 0.12 44 mg/L [2-5].” It is not clear which artificial sweetner(s) was found and which aquatic environments were analysed. To improve the comprehension of the introduction, I would suggest to the authors to add the missing information to this sentence.

1.3. In this manuscript the authors mention that the sweeteners concentration of 100ppb correspond to the “highest environmental relevant concentration”. I would suggest to the authors to further explain why 100ppb is the highest environmental relevant concentration.

2. Materials and methods

2.1. In Figure 1, the authors show the experimental design used in this work. To improve the reader comprehension, I would suggest to the authors to further detail the description of the experimental design in the legend p.e..

3. Results

3.1. The sentences from lines 169-179 correspond to background information. I would suggest to the authors to replace the sentence from line 169-179 from the results to introduction section.

3.2. The sentences in lines 180-182 and 196-199 describe the methodology used. I would suggest to the authors to place these sentences in methods section.

3.3. The text from lines 196-203 is repeated in lines 213-220

4. Appendix

4.1. The figure 1 in appendix seems to be missed in the manuscript text.

Author Response

In this manuscript, Saputra et al., showed the impact of acute and sub chronic exposure to 8 artificial sweeteners in zebrafish cardiac performance, by measuring cardiac physiological parameters. The authors showed that the studied sweeteners have a very little effect on zebrafish larva cardiac performance. I just have some minor suggestions to the authors to improve the manuscript

Minor concerns:

  1. Introduction

1.1. In the first paragraph of the manuscript, the authors start the introduction by describing the presence of artificial sweeteners in aquatic environment explaining their release, referring their concentration and difficulty in their degradation. On the second paragraph, the authors refer the consequences of the usage of artificial sweeteners in human diseases as obesity, diabetes and cardiovascular diseases showing also the data obtained from animal models. The relation between these two topics as well as the relation between the presence of artificial sweeteners in water and the aims of the present manuscript are not clear. I would suggest to the authors to add the missing information to introduction

Thank you for the valuable inputs. The author also agrees that the link between the first and the second paragraph should be included to increase reader comprehension. Therefore, in the updated manuscript, the authors already added the possibility of artificial sweeteners to be taken up by human.

1.2. In the sentence “Based on several previous study, artificial sweetener was found in several aquatic environment with concentration ranging from 3.5 ng/L to 0.12 44 mg/L [2-5].” It is not clear which artificial sweetner(s) was found and which aquatic environments were analysed. To improve the comprehension of the introduction, I would suggest to the authors to add the missing information to this sentence.

Thank you for the valuable suggestion. In line with the reviewer suggestion, the authors also think that adding the extra information will improve the reader comprehension. Thus in the updated manuscript, the information of the related artificial sweetener has been added.

1.3. In this manuscript the authors mention that the sweeteners concentration of 100ppb correspond to the “highest environmental relevant concentration”. I would suggest to the authors to further explain why 100ppb is the highest environmental relevant concentration.

Thank you for the suggestion. As in the first paragraph already mention, the highest environmental concentration of artificial sweetener reported by Gan et al. was 0.12 mg/L which is equivalent to 120 ppb then it was rounded down to 100 ppb. The authors strongly agree that this information should be included in the manuscript. Thus the authors already add the explanation in the updated manuscript.

  1. Materials and methods

2.1. In Figure 1, the authors show the experimental design used in this work. To improve the reader comprehension, I would suggest to the authors to further detail the description of the experimental design in the legend p.e..

Thank you for the valuable suggestion. The authors strongly agree that the detailed experimental design should be added in the manuscript. However, the authors believe that it will be better to add that in the material and method part section 2.1 rather than in the figure legend which have a very limited space. Thus in the updated manuscript, the authors already added the detailed protocol. Thank you for your professional comments.    

  1. Results

3.1. The sentences from lines 169-179 correspond to background information. I would suggest to the authors to replace the sentence from line 169-179 from the results to introduction section.

Thank you for the suggestion. The authors strongly agree that the sentences were correspond to background information. Thus in the updated manuscript, the sentences were already moved to the introduction section.

3.2. The sentences in lines 180-182 and 196-199 describe the methodology used. I would suggest to the authors to place these sentences in methods section.

The Authors thank you for the valuable suggestion. The authors also agree that the sentence in both mentioned lines were more fit to be placed in methods section, thus in the updated manuscript, both sentences were already moved.

3.3. The text from lines 196-203 is repeated in lines 213-220

Thank you for the correction. The updated manuscript has been change according to the reviewer correction.

  1. Appendix

4.1. The figure 1 in appendix seems to be missed in the manuscript text.

Thank you for the correction. The authors already add the missing part in the updated manuscript.

Reviewer 3 Report

None.

Author Response

Thanks

Reviewer 4 Report

I have appreciated the efforts of authors in modifying paper, nevertheless I think it is a low-medium level of manuscript with not so high interest for readers and especially practice.

Summing up, for me after modifications it might be suitable for considering acceptance by Editors and I'd like to cross final decision to them in order to satisfy better scientific purposes.

Author Response

I have appreciated the efforts of authors in modifying paper, nevertheless I think it is a low-medium level of manuscript with not so high interest for readers and especially practice. Summing up, for me after modifications it might be suitable for considering acceptance by Editors and I'd like to cross final decision to them in order to satisfy better scientific purposes.

Once again, the authors thank you for the thorough review by reviewer.

This manuscript is a resubmission of an earlier submission. The following is a list of the peer review reports and author responses from that submission.

Round 1

Reviewer 1 Report

In the paper named “low dose artificial sweeteners exposure can induce cardiac physiology alteration in Zebrafish” authors made a systematic study to explore the potential adverse effects of eight commercial artificial sweeteners. This study proves the use of Zebrafish as an “in vivo” model to evaluate the adverse effects of these artificial sweeteners in the cardiovascular systems.

There are only minor points.

  • Have authors tested other concentrations different from 100ppb or 100ppm?
  • Authors must provide some more date about the PCA and heatmap conditions in material and methods section.
  • In the same section, in point 2.3 the authors must give information about the data normalization type.
  • Between lines 163 and 181 authors shows data about heart variability, but there is not any figure or table showing this data.
  • In the manuscript there is some disorder in the figure sequence that makes difficult to follow the course of the results.
  • In line 192 the authors say that in figure 2D there is a decrease in beat after exposition to acesulfame K neotame, saccharine and sucralose values, but this is difficult to see. It is possible to change the bars scale to make this easier to see?
  • On the other hand in figure 2D authors say that no significant changes were observed but the graph shown great differences between bars.
  • In figure 3A the author states that there is a significant increase however in the sucralose there are not an asterisk of significance.
  • The majority of the figures are blurred and in some cases is difficult to see the significance asterisk.
  • In line 241 authors give us fig 4A and 4B as a reference of the data described but there is a mistake and the correct reference are figure 4A and 4C. The same happens in line 243 the author says figure 4C and D and should say 4B and D.
  • In point 3.4 (result section) author gives data about ROS levels. Can author explain how they obtain this data? This information is missing in the paper.

Author Response

In the paper named “low dose artificial sweeteners exposure can induce cardiac physiology alteration in Zebrafish” authors made a systematic study to explore the potential adverse effects of eight commercial artificial sweeteners. This study proves the use of Zebrafish as an “in vivo” model to evaluate the adverse effects of these artificial sweeteners in the cardiovascular systems.

There are only minor points.

  • Have authors tested other concentrations different from 100ppb or 100ppm?

Thank you for the comment. For pretesting, the authors tried several different concentrations ranging from environmental relevant concentration of 1 ppb to 10 ppm which is more than 10 times of the highest concentration found in environmental and found that 100 ppb is the lowest dose of artificial sweetener that already give significant result. On the other hand, the authors did not try concentration above 10 ppm because it is already way too high from the environmental relevant concentration (Gan, Sun, Feng, Wang, & Zhang, 2013; Kobetičová, Mocová, MrháLKová, Fryčová, & Kočí, 2016; Lange, Scheurer, & Brauch, 2012).

Gan, Z., Sun, H., Feng, B., Wang, R., & Zhang, Y. (2013). Occurrence of seven artificial sweeteners in the aquatic environment and precipitation of Tianjin, China. Water Research, 47(14), 4928-4937.

Kobetičová, K., Mocová, K. A., MrháLKová, L., Fryčová, Z., & Kočí, V. (2016). Artificial sweeteners and the environment. Czech Journal of Food Sciences, 34(2), 149-153.

Lange, F. T., Scheurer, M., & Brauch, H.-J. (2012). Artificial sweeteners—a recently recognized class of emerging environmental contaminants: a review. Analytical and bioanalytical chemistry, 403(9), 2503-2518.

  • Authors must provide some more date about the PCA and heatmap conditions in material and methods section.

Thank you for the suggestion. The authors also agree that instead of default it will be better to show the selected parameter for PCA and heatmap condition, thus the manuscript already updated according to the suggestion.

  • In the same section, in point 2.3 the authors must give information about the data normalization type.

The authors very thankful for the suggestion. The authors also agree with the reviewer that the information of normalization method should also described in the manuscript, thus the information of data normalization was added in the revised manuscript.

  • Between lines 163 and 181 authors shows data about heart variability, but there is not any figure or table showing this data.

Thank you for the thorough review. The heart rate variability described in line 163-181 was refer to heart regularity in figure 4. The authors also agree that this might make some confusion in the future, thus some change already done in the updated manuscript to address this problem.

  • In the manuscript there is some disorder in the figure sequence that makes difficult to follow the course of the results.

The authors thank you for the comment. The authors also agree that the arrangement of the manuscript especially in result part is a bit confusing, thus some change has been done in the revised manuscript.

  • In line 192 the authors say that in figure 2D there is a decrease in beat after exposition to acesulfame K neotame, saccharine and sucralose values, but this is difficult to see. It is possible to change the bars scale to make this easier to see?

Thank you for the valuable suggestion. In agreement with the reviewer suggestion, the decrease in figure 2D was difficult to see even after changing the scale bar, thus in the revised manuscript, the data for figure 2D and 3D has been deleted and the result part that discuss about it has been combined with 2C and 3C as both of talks about similar thing.

  • On the other hand in figure 2D authors say that no significant changes were observed but the graph shown great differences between bars.

Thank you for the comment. Please correct the authors if this was wrong but If this comment refer to figure 2E, the reason might be because of the data distribution that not following the normal distribution pattern. Thus, even if the mean value of the data is quite different, it is not significant if we do non-parametric test such as Kruskal-Wallis ANOVA test because non-parametric test will reduce the power of statistics (Colquhoun, 1971; Ghasemi & Zahediasl, 2012; Mishra et al., 2019).

Colquhoun, D. (1971). Lectures on Biostatistics: An Introduction to Statistics With Applications in Biology and Medicine. London, England: Clarendon Press.

Ghasemi, A., & Zahediasl, S. (2012). Normality tests for statistical analysis: a guide for non-statisticians. International journal of endocrinology and metabolism, 10(2), 486.

Mishra, P., Pandey, C. M., Singh, U., Gupta, A., Sahu, C., & Keshri, A. (2019). Descriptive statistics and normality tests for statistical data. Annals of cardiac anaesthesia, 22(1), 67.

  • In figure 3A the author states that there is a significant increase however in the sucralose there are not an asterisk of significance.

Thank you for the correction. We already corrected the manuscript to address this problem.

  • The majority of the figures are blurred and in some cases is difficult to see the significance asterisk.

Thank you for the valuable comment. The authors strongly agree that the significance asterisk is smaller than the authors intended, so the authors already change all the figure asterisk size to make it more visible.

  • In line 241 authors give us fig 4A and 4B as a reference of the data described but there is a mistake and the correct reference are figure 4A and 4C. The same happens in line 243 the author says figure 4C and D and should say 4B and D.

Thank you for the thorough review. Some correction has been done in the updated manuscript.

  • In point 3.4 (result section) author gives data about ROS levels. Can author explain how they obtain this data? This information is missing in the paper.

Thank you for the comment. The data for ROS was obtained from the mean value of ROS data which refer to figure 5D. The ROS concentration measurement in this study was done by using ELISA Kit according to the manufacturer information and the ROS measured was every superoxide anion radical (O2-), hydroxyl radical (OH-), peroxide groups (ROO-), hydroperoxy radical (HOO), nitric oxide (NO-), peroxynitrite anion (ONOO-), Hypochlorous acid (HOCl), semiquinone radical, singlet oxygen, and other intracellular reactive oxygen species superoxide group that present in the cell tissue suspension. Some slight increase can be seen according to the ELISA data, but it was not statistically significant which is according to the P value of the multiple comparison test. Based on this comment, the authors think that the conclusion of result part might confused the reader, so in the revised manuscript, the conclusion have been toned down to only address the significant change. 

Reviewer 2 Report

In this manuscript, Saputra et al., showed the impact of 8 artificial sweeteners in cardiac performance, by measuring cardiac physiological parameters in zebrafish larva exposed to these 8 compounds. The authors showed that saccharine and sucralose have a deleterious effect on zebrafish larva cardiac performance.

Minor concerns:

1. To facilitate the understanding of table 1, I would suggest to the authors to provide the information about which sweetener(s) were associated with each of the described diseases (“Related diseases” column) and with each biomarker change (“Effect on biomarker” column).

2. In lines 184, 204, 210 (as example) the authors mention a comparison of physiological parameters of animals exposed to sweeteners with a control group. In materials and methods, the authors explained what was used as positive and negative controls (line 102), however it is not clear what is the control group used for comparisons in line 184, 204 and 210. I suggest to the authors to clarify what is this control group.

3. To facilitate the reader comprehension, in figures 2, 3, 4 and 5, the authors should indicate the number of animals (N) used in the analysis, for example in figure legend.

4. In some figures the “*” indicating the statistical significant result is missing:

                4.1 In line 200, the authors mention that the larva exposure to sucralose at 10 ppm significantly changes the stroke volume. However, figure 3A does not have the significant “*” in the bar corresponding to this compound.

                4.2 In line 240 the authors wrote that acesulfame K significantly increase sd1, but the figure 4A does not have “*” in this compound.

5. I would suggest to the authors to tone down the conclusions to what is statistically significant different from the control group, as example:

                 5.1. In line 257, “Although we also observed some increase in AChE and cortisol after incubation in saccharine, it is not statistically significant (Fig. 5A & B). The slight increase also showed in ROS level after incubation in saccharine but did not reach statistic significant (P=0.107) (Fig. 5D).

                  5.2. In line 241, “While slightly elevation of heart rate variability after incubation in 10 ppm of aspartame was observed; however, not reach statistical significance (Fig. 4C & D).”

6. In discussion, line 367, the authors mention that “The positive effect of dopamine on heart rate and muscle contractility has been observed in animal models, such as dogs [85] and rats [86], thus corroborating our results that some artificial sweeteners can increase heart rate”, relating the dopamine with heart rate. However, in the present manuscript, the dopamine levels are only significantly different from controls after saccharine exposure and not after the exposure to the other sweeteners including sucralose, that led to alterations in heart rate but not in dopamine. Thus, the authors should rephrase this sentence.

7. In line 360, the authors mention that “After incubation in acesulfame K, neotame, saccharine, and sucralose, heart rate was significantly elevated in zebrafish.” The authors should clarify that this sentence refers to the exposure to these compounds at 100ppb (low concentration) and not at 10 ppm (high concentration). The authors should give a brief explanation why sucralose and saccharine increase the heart beat in larva exposed to these compounds at 10ppm and 100ppb whereas acesulfame K and neotame only show significant differences at 10ppb (low concentrations) and not at higher concentrations.

Author Response

Comments and Suggestions for Author

In this manuscript, Saputra et al., showed the impact of 8 artificial sweeteners in cardiac performance, by measuring cardiac physiological parameters in zebrafish larva exposed to these 8 compounds. The authors showed that saccharine and sucralose have a deleterious effect on zebrafish larva cardiac performance.

Minor concerns:

  1. To facilitate the understanding of table 1, I would suggest to the authors to provide the information about which sweetener(s) were associated with each of the described diseases (“Related diseases” column) and with each biomarker change (“Effect on biomarker” column).

Thank you for the suggestion. The authors strongly agree that addition of associated artificial sweetener makes table 1 easier to understand, thus the manuscript was already revised according to the suggestion.

  1. In lines 184, 204, 210 (as example) the authors mention a comparison of physiological parameters of animals exposed to sweeteners with a control group. In materials and methods, the authors explained what was used as positive and negative controls (line 102), however it is not clear what is the control group used for comparisons in line 184, 204 and 210. I suggest to the authors to clarify what is this control group.

Thank you for the suggestion. The control that noted in line 184, 204, and 210 was refer to non-treated group. The authors agree that this might confused the reader, thus the updated manuscript was revised to accommodate this problem.

  1. To facilitate the reader comprehension, in figures 2, 3, 4 and 5, the authors should indicate the number of animals (N) used in the analysis, for example in figure legend.

Thank you for the valuable suggestion. The number of animal used was already indicated in the material and method section, but the authors agree that the addition in figure legend will make it easier for the reader comprehension, thus the manuscript was updated according to the suggestion

  1. In some figures the “*” indicating the statistical significant result is missing:

4.1 In line 200, the authors mention that the larva exposure to sucralose at 10 ppm significantly changes the stroke volume. However, figure 3A does not have the significant “*” in the bar corresponding to this compound.

The authors thank you for the correction. The manuscript was corrected to accommodate this problem

4.2 In line 240 the authors wrote that acesulfame K significantly increase sd1, but the figure 4A does not have “*” in this compound.

Thank you for the correction. Some correction has been done in the updated manuscript

  1. I would suggest to the authors to tone down the conclusions to what is statistically significant different from the control group, as example:

5.1. In line 257, “Although we also observed some increase in AChE and cortisol after incubation in saccharine, it is not statistically significant (Fig. 5A & B). The slight increase also showed in ROS level after incubation in saccharine but did not reach statistic significant (P=0.107) (Fig. 5D).

Thank you for the valuable suggestion. The idea of highlight the slight alteration was to give the information to the reader about the compound that was almost reach the statistically significant. We also agree that some tone down was needed for the conclusions thus we revised the manuscript according to the suggestion.

5.2. In line 241, “While slightly elevation of heart rate variability after incubation in 10 ppm of aspartame was observed; however, not reach statistical significance (Fig. 4C & D).”

The manuscript was updated according to the suggestion.

  1. In discussion, line 367, the authors mention that “The positive effect of dopamine on heart rate and muscle contractility has been observed in animal models, such as dogs [85] and rats [86], thus corroborating our results that some artificial sweeteners can increase heart rate”, relating the dopamine with heart rate. However, in the present manuscript, the dopamine levels are only significantly different from controls after saccharine exposure and not after the exposure to the other sweeteners including sucralose that led to alterations in heart rate but not in dopamine. Thus, the authors should rephrase this sentence.

Thank you for the suggestion. The sentences were already rephrase in the updated manuscript.

  1. In line 360, the authors mention that “After incubation in acesulfame K, neotame, saccharine, and sucralose, heart rate was significantly elevated in zebrafish.” The authors should clarify that this sentence refers to the exposure to these compounds at 100ppb (low concentration) and not at 10 ppm (high concentration). The authors should give a brief explanation why sucralose and saccharine increase the heart beat in larva exposed to these compounds at 10ppm and 100ppb whereas acesulfame K and neotame only show significant differences at 10ppb (low concentrations) and not at higher concentrations.

Thank you for the comment. The authors also agree that the referred concentration also should be noted in in the manuscript. Thus, some change has been done in the revised manuscript. Furthermore, the possible mechanism why the significant change observed in incubation of 100 ppb of acesulfame K and neotame did not show at 10 ppm concentration was already added in the revised manuscript as the authors also agree that this explanation was needed by the reader.

Reviewer 3 Report

The manuscript by Saptura and coauthors entitled “Low Dose Artificial Sweeteners Exposure Can Induce Cardiac  Physiology Alteration in Zebrafish evaluated the cardiac performance and stress biomarkers in zebrafish embryos.

The results of this study demonstrated that none of the artificial sweeteners induce any embryonic death or cardiac deformation but showed alterations in cardiac physiological parameters with some of the artificial sweetners. Markers of stress remain unaltered except an increase in dopamine with Saccharine.

Title

The title of the study should include ‘acute’ exposure as this is a acute exposure study.

Abstract

Line#19: paraphrase this line to ‘an overdose of artificial sweeteners could result in adverse affects’. No need to mention after consumption as it is quite obvious.

Line#22: paraphrase this line to ‘on cardiac performances using zebrafish (Danio rerio) as a model system’.

Line 31-32: Instead of emphasizing the rationale of using zebrafish, authors are advised to summarize the results. This paper is not on comparing different model systems. Concluding line of the abstract should mention the main outcome of the study. Also, platform word is inappropriate. Instead, use model system or just model.

Introduction

Line 71-76: Complex sentence. Authors are advised to break in smaller sentences for improved comprehension.

Material and method

Line#90: heart beat in the (avoid using at the).

Line#96-100: Authors are advised to paraphrase these sentences for improved comprehension.  

Results

Line#163: start the sentence as ‘none of the artificial sweetners…..’

It is advisable to provide images showing no developmental defect in embryos such as edema.

Line#172: delete the word ‘can be’. Instead write ‘is divided into…’

Figure 5: why only lower does i.e. 100ppb is used in this experiment. Provide rationale.

Authors are advised to give at least the important pharmacokinetic parameters such as absorption, bioavailability, distribution, etc preferably in the form of table for each artificial sweetener used in the study. These are important characteristics for any compound testing and would be quite useful while interpreting the results and drawing conclusions.

Author Response

Comments and Suggestions for Authors

The manuscript by Saptura and coauthors entitled “Low Dose Artificial Sweeteners Exposure Can Induce Cardiac Physiology Alteration in Zebrafish evaluated the cardiac performance and stress biomarkers in zebrafish embryos. The results of this study demonstrated that none of the artificial sweeteners induce any embryonic death or cardiac deformation but showed alterations in cardiac physiological parameters with some of the artificial sweetners. Markers of stress remain unaltered except an increase in dopamine with Saccharine.

Title

The title of the study should include ‘acute’ exposure as this is a acute exposure study.

Thank you for the suggestion. The authors also agree with the reviewer that the tittle should include “Acute” to emphasize that this manuscript address the acute study only, thus the authors already change the manuscript tittle.

Abstract

Line#19: paraphrase this line to ‘an overdose of artificial sweeteners could result in adverse affects’. No need to mention after consumption as it is quite obvious.

Thank you for the suggestion. The authors also agree that line 19 should be paraphrase, thus some change already done according to the reviewer suggestion.

Line#22: paraphrase this line to ‘on cardiac performances using zebrafish (Danio rerio) as a model system’.

Thank you for the suggestion. In agreement with the reviewer suggestion, Line 22 already paraphrase according reviewer suggestion.

Line 31-32: Instead of emphasizing the rationale of using zebrafish, authors are advised to summarize the results. This paper is not on comparing different model systems. Concluding line of the abstract should mention the main outcome of the study. Also, platform word is inappropriate. Instead, use model system or just model.

The authors thank you for the valuable suggestion. The authors also agree with the reviewer that the conclusion in the abstract should emphasize the main outcome of the study, thus this part has been revised according to reviewer suggestion.

Introduction

Line 71-76: Complex sentence. Authors are advised to break in smaller sentences for improved comprehension.

Thank you for the suggestion. The authors strongly agree that line 71-76 consist of one long sentence that need to separate to improve the understanding of our intention, thus the updated manuscript was already updated according to the reviewer suggestion.

Material and method

Line#90: heart beat in the (avoid using at the).

Thank you for the correction. The authors already update the manuscript.

Line#96-100: Authors are advised to paraphrase these sentences for improved comprehension.  

Thank you for the suggestion. The authors also agree that line 96 - 100 need to be paraphrased to improve the understanding of our intention, thus we already updated the manuscript to accommodate this problem.

Results

Line#163: start the sentence as ‘none of the artificial sweetners…..’

Thank you for the correction. The updated manuscript has been revised according to the reviewer suggestion.

It is advisable to provide images showing no developmental defect in embryos such as edema.

Thank you for the advice. As the study focused on the cardiac physiology, the video that show no cardiac developmental defect especially edema has been included as supplementary video which focused on the heart chamber. However, in agreement with the reviewer, the images that show overall development of zebrafish embryo after artificial sweeteners exposure should also be provided, thus we already add the images as Appendix Figure A1 data in the revised manuscript.

Line#172: delete the word ‘can be’. Instead write ‘is divided into…’

Thank you for the correction. The sentence already paraphrase according to reviewer suggestion.

Figure 5: why only lower does i.e. 100ppb is used in this experiment. Provide rationale.

Thank you for the comment. The original idea of this study was to explore the adverse effect of various artificial sweetener at environmental relevant concentration. The concentration of 100 ppb was selected because that concentration was the lowest concentration that can alter the cardiac parameter we tested and still in the range of environmental relevant concentration (Gan et al., 2013; Kobetičová et al., 2016; Lange et al., 2012). The 10 ppm concentration was not included in biomarker testing because the authors think that 100 ppb already good enough to show the adverse effect of artificial sweetener and as the tittle explicitly tell, the authors want to explore the effect of low dose exposure to the cardiac parameter of zebrafish larvae. However, recent study showed that exposure of artificial sweetener at high concentration will have different biomarker profile compared to the lower dose exposure (Han, Li, Dong, Zhang, Gao, Li, & Du, 2021) which can be an interesting subject for future studies.  

Gan, Z., Sun, H., Feng, B., Wang, R., & Zhang, Y. (2013). Occurrence of seven artificial sweeteners in the aquatic environment and precipitation of Tianjin, China. Water Research, 47(14), 4928-4937.

Han, G., Li, X., Dong, G., Zhang, L., Gao, J., Li, M., & Du, L. (2021). Phenotyping Aquatic Neurotoxicity Induced by the Artificial Sweetener Saccharin at Sublethal Concentration Levels. Journal of Agricultural and Food Chemistry, 69(7) 2041-2050.

Kobetičová, K., Mocová, K. A., MrháLKová, L., Fryčová, Z., & Kočí, V. (2016). Artificial sweeteners and the environment. Czech Journal of Food Sciences, 34(2), 149-153.

Lange, F. T., Scheurer, M., & Brauch, H.-J. (2012). Artificial sweeteners—a recently recognized class of emerging environmental contaminants: a review. Analytical and bioanalytical chemistry, 403(9), 2503-2518.

Authors are advised to give at least the important pharmacokinetic parameters such as absorption, bioavailability, distribution, etc preferably in the form of table for each artificial sweetener used in the study. These are important characteristics for any compound testing and would be quite useful while interpreting the results and drawing conclusions.

Thank you for the valuable suggestion. Although as already briefly discussed in point 4.1 that most of the artificial sweeteners either poorly absorbed or rapidly metabolized, but the authors strongly agree that the addition of pharmacokinetics parameter will be useful to help the reader interpreting the results more clearly, thus in the updated manuscript, the authors already add new table that show some information from previous study about the absorption and metabolic fate of artificial sweeteners in human body.

Reviewer 4 Report

I think manuscript is globally quite good.

Minor spelling checks for English is required such as a minimal check of references because they are not always in accordance with journal's rules.

Nevertheless, I think it is suitable for publication with a grade of priority chosen by Editors because of very particular and/or exclusive topic.

Author Response

Comments and Suggestions for Authors

I think manuscript is globally quite good. Minor spelling checks for English is required such as a minimal check of references because they are not always in accordance with journal's rules. Nevertheless, I think it is suitable for publication with a grade of priority chosen by Editors because of very particular and/or exclusive topic.

Thank you for the encouraging comment. As the reviewer suggest, the authors already tried to do spelling checks as much as it can be for the updated manuscript.